# Channels and Transporters of the Pulmonary Lamellar Body in Health and Disease

**DOI:** 10.3390/cells11010045

**Published:** 2021-12-24

**Authors:** Paul Dietl, Manfred Frick

**Affiliations:** Institute of General Physiology, Ulm University, Albert-Einstein-Allee 11, 89081 Ulm, Germany

**Keywords:** lysosome related organelle (LRO), surfactant, alveolus, exocytosis, purinergic signaling, ivermectin, ambroxol

## Abstract

The lamellar body (LB) of the alveolar type II (ATII) cell is a lysosome-related organelle (LRO) that contains surfactant, a complex mix of mainly lipids and specific surfactant proteins. The major function of surfactant in the lung is the reduction of surface tension and stabilization of alveoli during respiration. Its lack or deficiency may cause various forms of respiratory distress syndrome (RDS). Surfactant is also part of the innate immune system in the lung, defending the organism against air-borne pathogens. The limiting (organelle) membrane that encloses the LB contains various transporters that are in part responsible for translocating lipids and other organic material into the LB. On the other hand, this membrane contains ion transporters and channels that maintain a specific internal ion composition including the acidic pH of about 5. Furthermore, P2X_4_ receptors, ligand gated ion channels of the danger signal ATP, are expressed in the limiting LB membrane. They play a role in boosting surfactant secretion and fluid clearance. In this review, we discuss the functions of these transporting pathways of the LB, including possible roles in disease and as therapeutic targets, including viral infections such as SARS-CoV-2.

## 1. Introduction

Lamellar bodies (LBs) are specialized lipid storage and secretory organelles found in various cells [1]. The most studied LBs are found in keratinocytes of the skin [2,3] and in alveolar type II (ATII) epithelial cells in the lung [4,5]. Next to lipids, LBs may also contain cell-type specific proteins and lytic enzymes. Due to their acidic pH and the shared compositional and physiological features with lysosomes, LBs have also been classified as lysosome-related organelles (LROs) [6,7].

LBs in ATII cells are storage organelles for pulmonary surfactant. Pulmonary surfactant is a complex mix of >90% lipids (mainly phospholipids) and specialized surfactant proteins. It is stored as densely packed multilamellar structures within LBs [8,9,10] and secreted into the alveolar lumen via exocytosis of LBs [5,11,12,13]. Secretion of pulmonary surfactant is essential for lung function. Defects in surfactant biogenesis or function are linked to a variety of severe diseases, some of which are directly related to perturbance of LB biogenesis and function [14,15,16,17].

The major function of pulmonary surfactant in the lung is the reduction of surface tension and stabilization of alveoli during respiration [18,19]. The surface tension in the lung creates a strong retractive force that tends to expel air and cause shrinkage of the organ. This was first discovered by von Neergaard in 1929 [20]. In the 1950s and early 1960s, a surface-active material (surfactant) that reduces the surface tension in the lungs was isolated and characterized by Clements [21,22]. By that time, LBs of the ATII cells were identified as the intracellular storage organelle for surfactant [4,23]. It was also shown that surfactant deficiency causes IRDS (infant respiratory distress syndrome), the first disease clearly related to immature LB biosynthesis in neonates that could be treated by surfactant replacement [24,25].

The first evidence in favor of an exocytotic release mechanism was provided by electron microscopy (EM) [26,27]. It has also been found that the phospholipid composition of LBs was similar to that of whole lung surfactant obtained from broncho–alveolar lavage (BAL) [28,29] and that LB exocytosis accounts for alveolar phospholipid composition [30].

Surfactant also contains four specific proteins (SP-A, SP-B, SP-C, and SP-D), which account for about 10% by weight [31]. They differ in their routes of secretion: The small hydrophobic SP-B and SP-C are localized within LBs and are co-secreted with phospholipids upon LB exocytosis. They are believed to play a role in enabling the formation of a highly organized, DPPC (dipalmitoyl phosphatidylcholine)-enriched, surface film [32]. The large, hydrophilic, surfactant proteins SP-A and SP-D, however, are secreted largely independently of LB contents. They are part of the collagenous family of proteins called collectins and appear to play an important role in host defense. SP-A and SP-D bind a wide spectrum of pathogens including viruses, bacteria, fungi, and pneumocystis (reviewed in [33]) fulfilling a crucial role in the innate pulmonary immune response, in addition to surfactant providing a physical barrier for pathogens, including severe acute respiratory syndrome coronavirus 2 (SARS-CoV-2) [34,35]. 

Many of the key features of alveolar LBs for surfactant maturation, including loading of lipids, maintenance of an acidic pH and intraorganellar ion homeostasis, as well as exocytosis and secretion of surfactant depend on functional ion channels and transport proteins localized on LBs. Despite this importance of the LB transportosome for LB homeostasis and function as well as for maintaining the vital surfactant system in the alveoli, our understanding thereof is far from completion. Within this review, we aim to summarize current knowledge on ion channel and transporter expression on LBs and their implication for LB function (physiology) and potential contribution to lung diseases (pathophysiology). 

## 2. Ion Channels and Transporters on Lamellar Bodies

Historically, interest in ion channels and transporters expressed on LBs in ATII cells was linked to their main function, the storage of surfactant lipids and proteins. Surfactant proteins, either freshly synthesized surfactant proteins B (SP-B) and C (SP-C) or surfactant proteins that are recycled from the alveolar space, reach the LBs through intracellular vesicle trafficking via multivesicular bodies or the endosomal recycling route, respectively [10,36,37]. Surfactant lipids, however, reach the LBs via non vesicular pathways. This requires lipid transfer proteins for intracellular lipid trafficking in ATII cells [38] and subsequently transporters for lipid translocation into LBs [10]. To date, several lipid translocation proteins have been identified on LBs by either immunofluorescence, immunohistochemistry, or Western blot from isolated LBs (Figure 1). These include ATP-binding cassette sub-family A member 3 (ABCA3) [39,40,41], lysosomal integral membrane protein-2 (LIMP-2, also known as SCARB2) [42], Niemann–Pick C1 (NPC1) and Niemann–Pick C2 (NPC2) [43], and P4-type ATPase ATP8A1 [44].

Channels and transporters on LBs are required to maintain the intra-organellar proton and ion concentrations, but also facilitate surfactant secretion following LB exocytosis. Yet, apart from the identification of various subunits of the vacuolar V-ATPase [45,46,47] only an outwardly directed Na^+^-K^+^-2Cl^-^ co-transporter (NKCC1 or SLC12A2, [45]), P2X_4_ purinergic receptors [41] and vesicular nucleotide transporter (VNUT or SLC17A9 [48]) have so far been identified on LBs and linked to a physiological function (Figure 1). Investigations of proton and electrical gradients also postulated the presence of Na^+^/H^+^ exchanger [45], Ca^2+^/H^+^ exchange(r), Ca^2+^-activated K^+^-channels, and possibly other K^+^-channels [49,50]. However, these have yet to be confirmed. 

In addition, proteomic studies have expanded the repertoire of proteins expressed on LBs [51,52,53]. These include additional lipid transporters (ABCA8a), amino acid transporters (SLC3A2), ion transporters (SLC4A1), or ATPases including Na^+^/K^+^ (ATP1A1) or Ca^2+^-transporting (ATP2B1, ATP2B4) ATPases. Most of these were exclusively detected in the limiting membrane fraction of isolated LBs. However, the relevance of these proteins for LB homeostasis and function is yet to determined. It also needs to be considered that many of the ion channels and transporters expressed on LBs are trafficked from the LBs to the plasma membrane during LB exocytosis and surfactant secretion, and can subsequently be either recycled back to LBs or sent to lysosomes for degradation [54,55]. For example, we have recently shown that P2X_4_ receptors that have been delivered to the cell surface upon LB exocytosis are recycled from the plasma membered back to LBs. This recycling to acidic organelles is required for re-sensitization of the receptors [55].

Additional ion channels and transporters have also been identified on other species of LROs [56] and future studies are required to confirm whether these are also present on LBs. It is not unlikely that LROs share a similar décor of transporter to support common features amongst LROs, however, some ion channels and transporters may be unique on individual LROs serving their specific function [53].

## 3. Physiological Role for Ion Channels and Transporters on Lamellar Bodies

Pulmonary surfactant is predominately composed of lipids (approximately 90%) with phospholipids being the most abundant ones (80% in mass). Cholesterol is also present at significant levels (10% in mass) [57]. Lipid loading into LBs depends on specific lipid transporters expressed on the limiting membrane of LBs. 

The best studied transporter is ABCA3 (reviewed in detail in [17]). Structure prediction and homology with other ABC transporters suggests that ABCA3 forms a lipid conduit channels in the membrane of LBs. The transfer of phospholipid species across the limiting membrane of LBs during their biogenesis is a highly energy-dependent process, catalyzed by binding and hydrolysis of ATP at two nucleotide-binding-domains [58,59,60]. It has been hypothesized that the energy accumulated during LB biogenesis confers an energy loaded state to the surfactant complexes, that converts LB particles (LBPs, i.e., the densely packed surfactant particles inside mature LBs) into a sort of pressurized particles, that readily transform into the surface-active interfacial film upon release from LBs [61,62] and contact with the air–liquid interface [63]. The detailed physiological substrates for ABCA3 are not yet fully characterized, but ample evidence suggests that ABCA3 facilitates loading of several phospholipid species into LBs. These include the main surfactant species phosphatidylcholine and phosphatidylglycerol, but also phosphatidylethanolamine and phosphatidylserine [60,64,65,66]. A role of ABCA3 in cholesterol loading is controversial [65,67]. 

LBs express cholesterol transporting LIMP-2 and the NPC2/NPC1 complex [42,43]. LIMP-2 and NPC1 transport cholesterol across the limiting membrane of lysosomes [68,69,70]. However, LIMP-2 and NPC1 shuttle cholesterol from the inside of lysosomes into the cytosol to facilitate cholesterol uptake into the cell. Hence cholesterol loading into LBs would require reverse orientation of either LIMP-2 or NPC1 in the LB membrane. Evidence against such inverted orientation of LIPM-2 comes from a recent study that suggested that LIMP-2 ferries the soluble enzyme peroxiredoxin 6 (PRDX6) to the lumen of LBs. This study further suggested that PRDX6 remains bound to a helical loop of LIMP-2 inside LBs. This loop is also localized on the luminal side in lysosomes, arguing against a reverse orientation of LIMP-2 in LBs [42]. Interestingly, PRDX6 exhibits phospholipase A2 (PLA2) activity in the acidic environment of the LB and plays a key role in LB phospholipid homeostasis [71,72]. PRDX6 facilitates the remodeling of unsaturated phosphatidylcholine (PC) to enrich desaturated phosphatidylcholine (DSPC) in pulmonary surfactant [73] but may also contribute to degradation of phospholipids [74]. Lack of PRDX6 is associated with increased LB phospholipid content [74,75]. It has been suggested that trapping PRDX6 to LIMP-2 in LBs facilitate interaction of PRDX6 with phospholipids that are shuttled through a hydrophobic tunnel in LIMP-2 [42,76]. The main function of LIMP-2 on LBs may therefore be linked to regulating LB phospholipid content rather than cholesterol loading into LBs. Similar results, that is, increases in LB phospholipid content, was observed in NPC1 and NPC2 mutant mice [77]. Moreover, LB cholesterol content was increased in ATII cells from these mice, arguing against a role of NPC1/NPC2 in cholesterol loading into LBs. Rather NPC1 (and LIMP-2) may be involved in extrusion of excess cholesterol from LBs that has accumulated during LB biogenesis. Accumulation could result from fusion of cholesterol containing lysosomes with multivesicular bodies (during LB biogenesis) or from cholesterol that is released by lipases from cholesteryl esters to release fatty acids [78]. In such a model, cholesterol is enriched and needs to be eliminated from the LB lumen to adjust surfactant cholesterol levels. Recently, it has also been suggested that increased vesicular cholesterol levels reduces fusion pore expansion and hence could limit secretion [79]. 

Whether ATP8A1, which is expressed on LBs, contributes to lipid loading into LBs or modulating surfactant lipid composition needs to be confirmed. ATP8A1 is a member of the Class P4 ATPases that flip phospholipids from one side of a membrane to the other using ATP hydrolysis as an energy source [80,81]. ATP8A1 has a high specificity for flipping phosphatidyl-serine (PS) across membranes [82]. Again, ATP8A1 normally flips PS from the luminal to the cytoplasmic leaflet of intracellular organelles [83], arguing against an active loading mechanism. Extrusion of excess PS accumulated during PB morphogenesis could be an alternative function, but again, no evidence so far supports such a mechanism. Recently it’s been discussed that a more likely role of ATP8A1 flippase activity is in preparation of LBs for exocytosis [44] similar to the observation that PS facilitates fusion of insulin-containing granules with the PM in pancreatic beta cells β and silencing of flippases impairs insulin secretion [84,85].

The lumen of LBs is acidic (pH of 5–5.5) [50], contains high concentrations of ionized Ca^2+^ [86,87], and has been proposed to only contain very little free water [61,88]. Establishing and maintaining such conditions requires transmembrane transport of ions and water. 

Acidification of isolated LBs depends on ATP and expression of the vacuolar V-ATPase on the LB membrane [45,46,50]. The acidic pH within LB is crucial for surfactant protein B and C processing [89,90,91] and the packaging of surfactant lipids [92]. In particular, the degradation and the remodeling of surfactant phospholipids by PRDX6, which expresses PLA2 and lysophosphatidylcholine acyl transferase (LPCAT) activities is strongly increased at acidic pH [71,93,94]. 

Besides the importance of intralumenal acidification for surfactant maturation, a shift in pH also affects intralumenal ion concentrations. It has been suggested that V-ATPase generates an electropositive LB lumen that ultimately affects the electrochemical gradient across the LB membrane via secondary active transport mechanisms. It´s been shown that the H^+^ influx is neutralized by a Cl^-^ uptake which in turn increases V-ATPase activity in a positive feedback loop [45]. This probably accounts for the Cl^-^ accumulation in LBs [86]. The entry of Cl^-^ could also reduce the K^+^-dependent electrical gradient contributing to the trans LB membrane potential. It´s also been proposed that a Na^+^/H^+^ exchanger dissociates the electrical and chemical H^+^ gradient exchanging H^+^ for Na^+^. The V-ATPase could also serve as the driving force for the outwardly directed Na^+^-K^+^-2Cl^-^ co-transporter [45].

An intimate link exists between acidic pH in LBs and luminal Ca^2+^ concentrations. Ca^2+^ uptake depends on the low pH in LBs [45]. Again, this is likely via secondary active transport of Ca^2+^ by a Ca^2+^/H^+^ exchanger [49]. Additional mechanism for Ca^2+^ uptake into LBs have been discussed [49] but may not be relevant at the very low cytoplasmic Ca^2+^ concentrations found in ATII cells [95,96]. Alkalinization of LBs, e.g., by inhibition of the vacuolar V-ATPase by bafilomycin A1 or H^+^ sequestration by ambroxol results in Ca^2+^ release from LBs and stimulates LB exocytosis and surfactant secretion from LBs [46,97]. The detailed pathways for this pH-dependent Ca^2+^ release are not yet fully understood. Two pore channels (TPCs), mucolipin TRP channels (TRPMLs), and P2X purinergic receptors have been linked to pH-dependent Ca^2+^ release from lysosomes [98,99,100]. We have recently reported that P2X_4_ receptors are expressed on LBs [41]. Alkalinization induced Ca^2+^ release from LBs through P2X_4_ receptors [48]. Activation of P2X_4_ receptors also depends on the presence of its natural ligand ATP. ATP is loaded into LBs by VNUT and present at high concentrations (∼1.9 mM) inside LBs [48]. The role for this alkalinization-induced, intracellular Ca^2+^ release is not yet clear. In lysosomes P2X_4_ forms channels activated by luminal ATP in a pH-dependent manner [101] and P2X_4_-mediated endolysosomal Ca^2+^ release promotes lysosome fusion [98]. Such fusion events have not been observed for mature LBs but may be relevant during LB biogenesis or in case of cell damage, when alkalization promotes a massive Ca^2+^ release from LBs to induce autophagy [102]. The most important function for P2X_4_ receptors on LBs, however, is to facilitate secretion and activation of pulmonary surfactant. Upon exocytosis of LBs and opening of the fusion pore, the intraluminal pH is rapidly neutralized and luminal ATP may activate the P2X_4_ receptors. This results in a fusion-activated cation entry (FACE) via P2X_4_ at the site of LB exocytosis. The resulting Ca^2+^ elevation around the fused LB accelerates the widening of the narrow fusion pore that restricts efficient release (secretion) of poorly soluble surfactant, i.e., LBPs [103,104,105,106]. This vital function is also supported by the observation that the Ca^2+^ concentration in exocytotic LBs is higher than in perinuclear LBs [86]. Moreover, FACE drives a net transepithelial fluid transport from the alveolar lumen. The resulting reduction of alveolar lining fluid thickness promotes direct contact between newly released surfactant and the air–liquid interphase, thus facilitating its adsorption and activation [107,108]. The multiple functions of FACE have been reviewed and represented in detail, recently [105,107,109].

Studies analyzing the structural organization of LBs propose a continuous intraorganellar dehydration during maturation of LBs as a result of the massive lipid accumulation by ABCA3 [61,88]. It´s been proposed that phospholipids adopt a particular high-energy structure as a result of the energy accumulated during continuous packing of lipids into the limited volume of LBs. The high level of dehydration might be promoted both by the high packing and as a consequence of osmotic stress induced by the segregation of large protein complexes out from the tightly packed multilamellar arrays [61]. Whether water is predominantly removed from tightly packed LBPs (but still residing inside the LB) or irrevocably extruded from the LB is yet to be determined. In the latter case water can either permeate passively across the LB membrane or via water conducting structures [110,111,112,113]. However, such entities (e.g., aquaporins) have not yet been described on LBs. A loss of free water would also affect intraorganellar pH and ion concentrations and raises the question if and how these are regulated. The observation that mature LBs contain high amounts of Ca^2+^ [86] is consistent with the general property of lysosomes as sites of intracellular Ca^2+^ storage [114] and in addition may result from concentration as a result of water loss.

## 4. Pathophysiology Linked to Ion Channels and Transporters on Lamellar Bodies

Defects in LB channel or transporter function are linked to a variety of diseases in the lung, including interstitial lung diseases (e.g., pulmonary fibrosis). Owing to its classification as a LRO, these channels and transporters are also intrinsically associated with functions required in host defense, including viral infections. Defects in lipid transporters affect surfactant maturation and LB homeostasis. This can result in changes in LB size and phospholipid content that leads to ATII cell injury and ultimately to development of pulmonary fibrosis. Damaged ATII cells secrete mediators that lead to fibroblast proliferation and differentiation to highly active myofibroblasts, which deposit excessive amounts of extracellular matrix (ECM) [115,116]. This results in overall remodeling of the alveolar structure, formation of scar tissue, thickening of the alveolar septae, and an increase in tissue stiffness [117,118].

The pathophysiology of the ABCA3 transporter is by far the best studied and has been outlined in extensive reviews elsewhere [17,119]. Over 200 distinct ABCA3 mutations have been identified. These constitute the most prevalent group of mutations among genes associated with surfactant-related lung disorders (excellently summarized in [17]). Mutations can either affect intracellular trafficking of ABCA3 (type I), ATP hydrolysis (type II) or both (i.e., heterozygote mutations, type III). Compound heterozygous variants appear to account for increasing disease severity. Mutations in the ABCA3 gene are associated with surfactant dysfunction [120,121], familial lung diseases ranging from respiratory failure in newborns or interstitial lung disease in children [122,123] to idiopathic pulmonary fibrosis (IPF) or diffuse parenchymal lung disease in adults [124,125,126]. Most ABCA3 mutations result in an ABCA3 null phenotype and are lethal within the first months following birth. Ultrastructural examination of lung tissues from these patients revealed a lack of mature LBs. Instead, numerous smaller vesicles with denser inclusion bodies were observed [10,65,127]. ABCA3 mutations that result in either partial loss-of-function or promote a toxic gain-of-function phenotype are less frequent and are associated with a more chronic disease phenotype.

LBs also contain transporters linked to lysosomal storage diseases [43,128]. Lysosomal storage diseases (LSDs) are inherited metabolic disorders characterized by the gradual accumulation of substrates inside lysosomes or LROs (that is, ‘storage’), which ultimately leads to cell dysfunction and cell death [129,130]. LSDs comprise a group of 70 monogenic disorders, several of which are associated with a pulmonary phenotype [130], ranging from the upper airways to the lung parenchyma (reviewed in detail in [131]). In this review, we consider diseases that are explicitly associated with a transporter found on LBs–Niemann–Pick disease (NPD) and Hermansky–Pudlak Syndrome (HPS).

NPD consist of autosomal recessive disorders associated with neurologic symptoms, splenomegaly, and the storage of lipids, including cholesterol. Three subtypes of the disease are described: Niemann-Pick disease type A, B and C. Niemann-Pick disease type C (NPC) is caused by mutations of the NPC1 and NPC2 genes that result in impaired cellular processing and transport of low-density lipoprotein (LDL)-cholesterol. NPC1 and NPC2 are found on LBs [132]. The majority (95%) of cases of NPC disease are caused by a mutation in NPC1, while only about 5% are due to mutations in NPC2 [43]. NPC1-mutant type II cells had uncharacteristically larger LB (mean area 2-fold larger), while NPC2-mutant cells had predominantly smaller LBs (mean area 50% of normal) than wild type [77]. The manifestation of NPD in the lung is an interstitial lung disease which is characterized by reduced diffusion capacity for carbon monoxide (DLCO) [130]. Pulmonary involvement in NPC1 was reported to be as well associated with an obstructive and restrictive lung disease [132]. However, to our knowledge, a causal link between a specific mutation and defective cholesterol transport in LBs to clinical manifestations (i.e., interstitial lung disease, fibrosis) has yet to be established. Recently, it has also been hypothesized that the inherent cellular and biochemical abnormalities of lysosomal storage diseases (LSDs) in general, and Niemann–Pick disease type C (NPC) in particular, create “unfavorable” (lysosomal) environments for SARS-CoV-2 infectivity in the host cells, that is ATII cells (see also below) [133].

HPS is a rare, genetic, multisystem disorder characterized by oculocutaneous albinism, bleeding diathesis, immunodeficiency, granulomatous colitis, and pulmonary fibrosis (reviewed in [16]). It represents a family of disorders in which the biogenesis of lysosome-related organelles (LROs) is compromised [134,135]. The underlying defects of HPS are mutations in genes that encode proteins which are essential for the synthesis of LROs, including the Biogenesis of LRO Complexes (BLOC)-1, -2, and -3 and the Adaptor Protein 3 complex (AP-3). Three subtypes of HPS, HPS1, -2, and -4, are associated with giant LBs, impaired surfactant secretion, ATII cell hyperplasia and fibrosis [136,137,138,139]. In ATII cells, it was recently shown that AP-3 is required to sort LIMP-2 and PRDX6 [42] but also the P4-type ATPase ATP8A1 from early endosomes to LBs [44]. Impaired trafficking of the luminal enzyme PRDX6 was associated with increased LB phospholipid content, that may lead to ATII cell injury and fibrosis [42]. ATP8A1 is a flippase that hydrolysis ATP to flip phospholipids from one side of a membrane to the other. Disruption of the AP-3/ATP8A1 interaction causes activation of Yes-activating protein, a transcriptional coactivator that augments cell migration and ATII cell numbers. It was suggested that this causes a toxic gain-of-function that results in activation of a repair process associated with severe, progressive pulmonary disease and fibrosis [44].

Much less is known about functional roles for ion channels or receptors on LBs in the pathogenesis of pulmonary diseases. Here we speculate whether LBs, and in particular P2X_4_ receptors expressed on LBs, may play a role in SARS-CoV-2 induced disease. Viruses such as SARS-CoV and SARS-CoV-2 are taken up by ATII cells after binding of the viral spike protein (S protein) to the SARS receptor, the angiotensin converting enzyme-2 (ACE2), on the host cell [140,141,142,143,144]. Entry of viral mRNA into the cytoplasm occurs by two possible mechanisms: First, by direct fusion of the viral membrane with the plasma membrane. This mechanism is mediated by transmembrane protease serine subtype 2 (TMPRSS2)-induced cleavage of the ACE2-spike protein (ACE2-S) complex [145,146,147]. ATII cells express both, ACE2 and TMPRSS2 [140,148,149,150,151]. Second, and more importantly, binding of S protein to ACE2 induces endocytosis of the virion. Cleavage of the viral proteins’ ‘S2′ site’ by cathepsin in the acidic endo-lysosomal compartment then induces fusion of the viral envelope with the late endosome/lysosome membrane to release the viral genome into the cytoplasm [152,153,154,155,156]. This is also referred to as “uncoating” [157,158]. 

As outlined above, LBs have a vesicular pH of about 5 [50,159,160] and also contain proteases such as cathepsins and others that belong to the class of lysosomal enzymes required for the viral uncoating [52,161,162]. During biogenesis LBs derive from late endosomes (multivesicular bodies) and receive cargo from the trans-Golgi network. In theory, LBs or fusion products of LBs with other acidic (i.e., endo-lysosomal) compartments could therefore constitute intracellular hubs for uncoating and processing of SARS-CoV-2 in the lung. In line, SARS-CoV-2 has been found in LBs of induced pluripotent stem cell-derived ATII (iATII) cells maintained at air–liquid culture conditions [163]. However, SARS-CoV and SARS-CoV-2 viral particles were not readily visible in LBs of infected primary ATII cells investigated by electron microscopy [143,164].

It is tempting to speculate whether LBs contribute to SARS-CoV-2 infection or clearance. There is ample evidence that lysosomotropic drugs and endosomal acidification inhibitors like (hydroxy)chloroquine, that are weak bases and trapped within acidic compartments, exert anti-viral activities. Through alkalinization of the vesicles, these compounds inhibit the activity of cathepsin, which is required for the viral uncoating process, [157,165,166]. We have recently demonstrated that ambroxol, another drug with anti-viral activities used to ameliorate respiratory infections, accumulates in LBs, and neutralizes LB pH by the same mechanism [97]. It has also been shown that ambroxol prevents SARS-CoV-2 entry into other epithelial cells by inhibition of acid sphingomyelinase [167]. As noted above, ambroxol also elicits Ca^2+^ release from LBs via alkalinization-dependent activation of P2X_4_ receptors on LBs. This results in LB fusion with the plasma membrane and surfactant release [97]. Thus, alkalinizing drugs are not only potential inhibitors of viral uncoating in LBs but could also contribute to viral clearance from ATII cells via stimulation of LB exocytosis and surfactant (content) secretion. 

P2X_4_ receptors expressed on LB membrane could also play an important role for viral inactivation and protection of the alveolus from SARS-CoV-2-induced alveolar barrier damage and acute respiratory distress syndrome (ARDS). ARDS by any cause, including SARS-CoV-2 infection, is accompanied by a loss of alveolar barrier function, surfactant deficiency or dysfunction, changes of alveolar compliance, hypoxia, and mechanical stress of alveolar units (reviewed in [168,169,170]). ARDS-associated cell damage and/or the activation of the immune response can lead to the release of cytokines, including ATP, from various cell types. ATP is a danger signal in the lung [171,172]. In healthy conditions, extracellular ATP within the respiratory lining fluid is continuously hydrolyzed by the action of ectonucleotidases [172] and it is intrinsically difficult to estimate the concentration of ATP in the alveolar hypophase between the surface layer of secreted surfactant and the apical membrane of ATI and ATII cells, since this liquid layer has an estimated mean thickness of about 200 nm only. However, strong ATP release from epithelial cells has been demonstrated for various ways of cell stress, including hypoxia or mechanical stress [173,174] and epithelial cell lysis induced by viral infection will inevitably lead to a strong increase of the ATP concentration in the adjacent pulmonary lining fluid. Accordingly, significant elevations of ATP concentrations have been demonstrated in the bronchoalveolar lavage (BAL) fluid under various pathophysiological conditions including SARS-CoV-2 infection [171,175]. This led to ATP-driven purinergic-inflammasome signaling [175]. In addition, ATP is released from LBs upon LB exocytosis [48], which can also lead to a (temporal) increase in extracellular ATP levels. The physiological significance of this last mechanism is not entirely clear but most likely part of a positive feedback mechanism enabling or facilitating the release of surfactant by expansion of the fusion pore. Although it is not entirely clear at which stage of the infection the ATP concentration starts to rise in the alveolar space, there is overwhelming plausibility that it can reach levels sufficient for activation of P2X_4_ receptors. P2X_4_ receptors on LBs are integrated into the apical membrane of ATII cells upon LB exocytosis and the ATP binding site of the P2X_4_ receptor is suddenly exposed to the alveolar lumen [41]. In the absence of ATP, this added apical membrane is quite “tight”, because even when cell membrane capacitance increases by more than 10%, the cell membrane conductance is unaltered [160]. When ATP is on the apical side, however, it can readily bind to the receptor, enabling the entry of cations (mainly Ca^2+^ and Na^+^ ions) from the alveolar hypophase (or edema fluid) into the cytoplasm of type II cells [41]. This has two consequences:

First, a Ca^2+^ signal is generated in the ATII cell, which we termed “fusion-activated Ca^2+^ entry”, because it follows the fusion of an LB with the plasma membrane (see above). FACE facilitates surfactant secretion from the fused LB, and activates LB exocytosis. Second, Na^+^ entry into the cytoplasm through the P2X_4_ receptor removes osmotically active Na^+^ from the alveolar lumen, facilitating alveolar clearance from excess edema fluid by osmosis [107]. Both mechanisms potentially protect the alveolus by restoring surface tension and alveolar compliance on the one hand, and by reducing fluid accumulation and improving oxygenation on the other hand. Last but not least, the release of surfactant as part of the innate immune system facilitates viral clearance in many ways, as reviewed in detail [176].

Interestingly, fusion-activated Ca^2+^ entry into type II pneumocytes is potentiated by ivermectin [41], an allosteric positive potentiator of P2X_4_ receptors [172,177,178]. Ivermectin, an antiparasitic drug used to treat persons or animals infested with helminths and insects, has gained substantial interest during the COVID-19 crisis, because it was used as an off-label drug to protect from COVID-19 infection. The issue is heavily debated and controversially discussed, as exemplified in a recent exchange of letters [179]. The general believes of a lack of plausibility that ivermectin may help to treat COVID-19 comes from its action on glutamate-gated chloride channels common to invertebrate nerve and muscle cells. These channels are not expressed in humans. Interestingly, the P2X_4_ receptor is never considered as a potential target of ivermectin in the COVID-19 literature. It should be considered that ivermectin is quite lipophilic, and lipophilic drugs can accumulate in the lipid environment of a LB [180]. For this reason, ivermectin may reach high local concentrations at the site of the P2X_4_ receptor when LBs are exocytosed. Given this experimental and theoretical background, ivermectin may be considered as an activator of the P2X_4_ receptor for a possible effect against COVID-19.

Apart from these speculative considerations regarding ivermectin, to date no specific treatment exists for disorders caused by defects in ion channels or transporters on LBs. This may be related to the fact that still little is known about the exact function of most of these proteins for LB biogenesis, homeostasis, a possible role in surfactant dysfunction or lung diseases related to ATII cell dysfunction. Even for ABCA3, which is clearly associated with development of various lung diseases, no specific treatment exists for disorders caused ABCA3 mutations [17]. One possible strategy to tackle this need could be a similar approach that has been very successful in the development of therapeutics for treatment of cystic fibrosis (CF). CF is caused by mutations of the cystic fibrosis transmembrane conductance regulator (CFTR) channel. Nearly 2000 cystic fibrosis-causing mutations have been described, many of which result in trafficking and / or channel conductance defects [181]. Development of small molecule CFTR modulators that either aid the trafficking (“correctors”) or improve the function (“potentiators”) of mutated CFTR or increase the amount of CFTR mRNA (and therefor protein, “amplifiers”) have been developed and are already approved drugs or entered clinical trials [182,183,184,185,186]. Such approaches (e.g., exploiting high-throughput screening technologies) may have the potential to identify novel drugs to correct mistrafficking and/or dysfunction of mutant ABCA3 isoforms. 

Alternatively, the future may also bring novel therapeutic options based on gene therapy to treatment of genetic diseases that are beyond the reach of traditional approaches. The goal of gene therapy for genetic diseases is to achieve durable expression of the therapeutic gene or “transgene” at a level sufficient to ameliorate or cure disease symptoms with minimal adverse events. Hundreds of gene-therapy programs are in clinical development and several gene therapy products have already been approved [187].

Last but not least, repurposing of approved drugs could be a swift way to receive authority approval for novel applications. Several drugs have been developed and approved for clinical use that specifically target some of the channels and transporters expressed on LBs, e.g., clodronate, an anti-osteoporotic drug that inhibits VNUT [188]. Other approved drugs have been found to affect some of these proteins rather unexpectedly. Ambroxol, an over-the-counter mucolytic drug has been found to increase the levels of LIMP-2 in neurons of mice [189]. Whether this is related to the effects of ambroxol on intra-organellar pH levels needs to be seen.

Overall, although there are currently no specific treatments available for disorders related to ion channels and transporters on LBs, the future holds multiple strategic options for development of such therapeutics. However, in addition to drug development efforts, success will also depend on a better understanding of the exact physiological function of the individual transport proteins on LBs and how misfunction contributes to disease development. 

## 5. Summary and Outlook

The pulmonary LB was originally discovered as the storage organelle for surfactant, the vital multi-molecular substance needed for breathing and preventing respiratory distress of the neonate [4]. Since then, our knowledge about the LB has been continuously expanding (Table 1), and its equipment as a LRO with a growing number of transporters, channels, and receptors prompts questions about its involvement in multiple additional functions/dysfunctions, including its exact role in the pathogenesis of chronic pulmonary disease such as fibrosis of various origin, and of its possible involvement in the processing and transmission of viral disease. 

## Figures and Tables

**Figure 1 cells-11-00045-f001:**
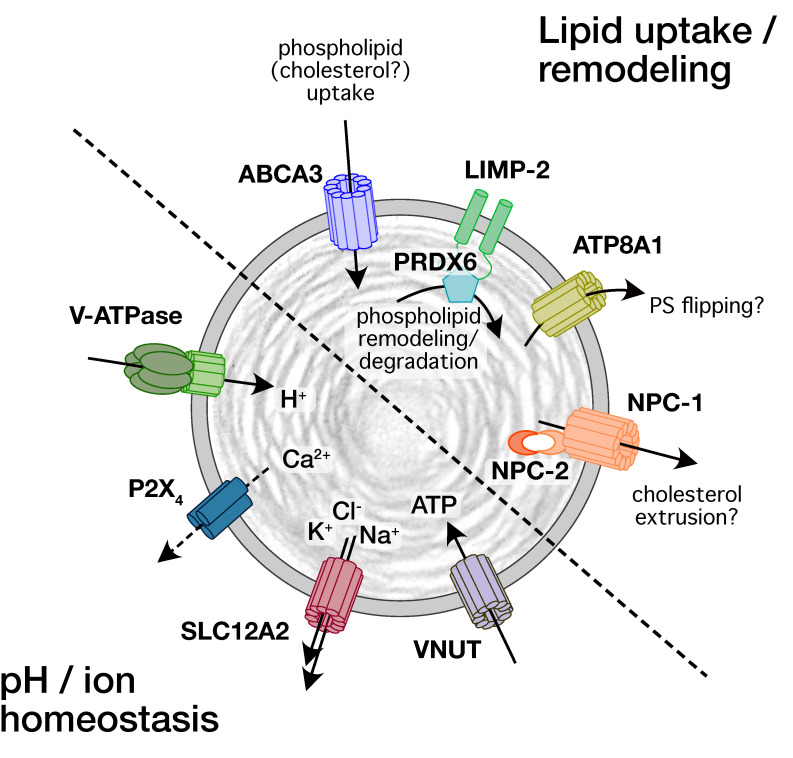
Schematic representation of transporters and channels identified on LBs and their proposed function for lipid and pH/ion homeostasis.

**Table 1 cells-11-00045-t001:** Channels and transporters that have been identified on LBs and for which a function and/or a possible relevance in lung disease has been reported. The table does not list channels and transporters identified on LBs for which no physiological function has been described so far.

Ion Channel Transporter	Detection	Physiological Function	Role in Lung Disease
ABCA3	Immuno-EM, IF [39]	LB biogenesis, lipid uptake [17,60,64]	Surfactant-related lung disorders [120,121] respiratory distress in newborns [122,123] interstitial lung diseases (ILDs), fibrosis [124,125,126]
ATP8A1	WB, IF [44]	Suggested: LB priming for exocytosis [44]	Possible involvement in fibrosis [44]
LIMP-2/SCARB2	WB, IF [42]	Possibly role in luminal localization of PRDX6 for regulation of LB phospholipid content [42]	n.d. (possibly fibrosis [42])
NPC1	WB, IF [43]	n.d. (possible role in regulating LB cholesterol content)	ILD, fibrosis [132,190]
NPC2	WB, IF [43]	n.d.	Fibrosis [190]
V-ATPase subunits	WB [45,46,47] IF [46,47]	Acidification of LB lumen [45,50]	n.d.
P2X_4_	WB, IF [41]	Ca^2+^ release/entry (FACE) facilitates surfactant secretion and activation, alveolar fluid resorption [41,107]	n.d.
SLC12A2	WB [45]	Na^+^, K^+^, 2Cl^-^ efflux [45]	n.d.
VNUT	WB, IF [48]	ATP uptake [48]	n.d.

Abbreviations: WB, Western blot of isolated/enriched LBs; IF, immune-fluorescence; immuno-EM, immune-gold staining in transmission electron microscopy.

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
