# Peer review of "Channels and Transporters of the Pulmonary Lamellar Body in Health and Disease"

_cells, 2021, doi:10.3390/cells11010045_

Round 1

Reviewer 1 Report

This Review summarizes current knowledge on ion channels and transporters on lamellar bodies of alveolar Type II cells and their implication for LB function in health and disease. It is, to my knowledge, the only publication covering this topic so comprehensively. Particular attention is payed to SARS-CoV2 infections, which makes this paper especially significant. The authors even suggest a previously unrecognized molecular mechanism how ivermectin might interfere with ATII cells/surfactant/LB homeostasis. I have only a few minor comments:

„The pulmonary lamellar body (LB) of the alveolar type II (ATII) pneumocyte....“ sounds strange (at least pulmonary, pneumocyte and alveolar is redundant. It’s clear, the paper is dealing with the  lung). Can that be improved? (abstract, line 8)

Wondering if the term “substance” (abstract, line 9) is correct, considering that surfactant is rather a mixture of different substances.

defending against (abstract, line 12)

SARS-CoV2: Use a consistent notation

Application/non-application of hyphenation in all headers is unclear

Maybe the authors should once explain the term “limiting membrane”

Labelling in Fig. 1 and 2 makes a sloppy impression.

pH (ph) in Legend to Fig. 1, line 88

Sentence mistake in line 98

Some statements lack a critical assessment, for example:

The lumem of LBs.... is almost completely dehydrated in mature LBS (line 152-153). I’m not aware of anything in the cell which is completely dehydrated. At least it is in contrast to their own findings (Fois et al. 2015) that Ambroxol and many soluble experimental agents show unhindered diffusion right into LBs, but how would that be possible into a dehydrated matter?

And lines 34-35 in Introduction: Evidence for facilitated O2 diffusion is also weak and based on a single paper and not shown in vivo, but reads here, again, as a generally accepted view.

Maybe these passages should be weakened, at least at first mentioning, since support is either weak, indirect or controversial. I propose that the authors carefully check all other chapters whether the referenced works support a consensus view, or merely an opinion.

line 336: ....has an estimated mean thickness of.....

Explain ILD in Table 1

Reviewer 2 Report

Summary

The review of Dietl and Frick describes the role of transporters, ion channels, and the P2X4 receptor in the regulation of LB functions. The review is concise and the text is clear and easy to read. However, I think that the review still misses a clear link with “Health and Disease” which is the focus of the current special issue. Thus, in order to be suitable for publication some edits are requested:

Major concerns:

-Some historical information regarding LB discovery and the composition of surfactant, including a brief explanation of how surfactant participates in the maintenance of respiratory tract functions, would help i) the narrative and ii) highlight the importance of channels and transporters described thereafter. In this context, some seminal references are missing (e.g. Balis & Connen 1964 - PMID: 14212352. NOTE that this is just an example but additional historical papers should be included).

-Paragraph 4 “Pathophysiology linked to Ion channels and Transporters”

the paragraph would benefit from:

  1. a table including the disease associated mutants for every transporter/ion channel described and clearly involved in LB function.
  2. a deeper description of the effect that a specific mutation has on LB function
  3. a section including the pharmacological modulation or alternative therapeutic strategies available, or/and those currently under investigation, of these targets. This part is very important owing the recent success of CFTR inhibitors in cystic fibrosis. A speculation of future pharmacological directions would also be beneficial to suggest future directions.

Please expand this section.

- Figure2:

I am not sure if in the current form Figure 2 can add something to the review. If the idea is to highlight the involvement of P2X4 in LB functions, the figure in the current status does not impress me. I suggest modifying the figure to highlight instead the different processes that could be impacted by P2X4 modulation. This is what, as a reader, I would expect.

Minor concerns:

-Abstract: “Surfactant reduces the surface tension in the alveolus and enables inspiration.” Isn’t the surfactant function going beyond the “simple” inspiration only? Doesn’t it contribute to the overall functionality of the lung?  

-The fact that LB are secretory machineries found exclusively in type II alveolar cells of the lung and keratinocytes of the skin should be highlighted.

-Line76-79. Specify if a functional consequence of such trafficking has been suggested

-Line 152: acidic but in which range? Having this number might help the reader in setting up experimental conditions.

-210-212: “In the latter case water can either permeate passively across the LB membrane or via water conducting structures [107–110], however, such entities (e.g., aquaporins) have not yet been described on LBs. “        please, split the sentence.

-Line 294: TMPRSS2 is not introduced (Transmembrane Serine Protease 2 ?)

-Line 349: substitute “values” with “levels”

-Figures 1 and 2: check that the fonts are consistent with the text. 

Round 2

Reviewer 2 Report

I have no further recommendations - the Authors have sufficiently satisfied my requests and I fully support the publication of their review.